# Structure–Function Relationship of a Novel MTX-like Peptide (MTX1) Isolated and Characterized from the Venom of the Scorpion *Maurus palmatus*

**DOI:** 10.3390/ijms251910472

**Published:** 2024-09-28

**Authors:** Rym ElFessi, Oussema Khamessi, Michel De Waard, Najet Srairi-Abid, Kais Ghedira, Riadh Marrouchi, Riadh Kharrat

**Affiliations:** 1Laboratory of Venoms and Therapeutic Biomolecules, Pasteur Institute of Tunis, University of Tunis El Manar, 13 Place Pasteur, BP74, Tunis 1002, Tunisia; rym.elfessi@pasteur.utm.tn (R.E.); oussama.khamassi@pasteur.tn (O.K.); riadh.marrouchi@pasteur.tn (R.M.); 2Laboratory of Bioinformatics, Biomathematics and Biostatistics (LR20IPT09), Pasteur Institute of Tunis, University of Tunis El Manar, Tunis 1002, Tunisia; kais.ghedira@pasteur.tn; 3l’Institut du Thorax, Nantes Université, Centre National de la Recherche Scientifique (CNRS), Institut National de la Santé Et de la Recherche Médical (INSERM), F-44000 Nantes, France; michel.dewaard@univ-nantes.fr; 4LR20IPT01 Biomolécules, Venins et Applications Théranostiques, Institut Pasteur de Tunis, Université de Tunis El Manar, Tunis 1002, Tunisia

**Keywords:** scorpion venom, maurotoxin, α-KTX, potassium channel

## Abstract

Maurotoxin (MTX) is a 34-residue peptide from *Scorpio maurus* venom. It is reticulated by four disulfide bridges with a unique arrangement compared to other scorpion toxins that target potassium (K^+^) channels. Structure–activity relationship studies have not been well performed for this toxin family. The screening of *Scorpio maurus* venom was performed by different steps of fractionation, followed by the ELISA test, using MTX antibodies, to isolate an MTX-like peptide. In vitro, in vivo and computational studies were performed to study the structure–activity relationship of the new isolated peptide. We isolated a new peptide designated MTX1, structurally related to MTX. It demonstrated toxicity on mice eight times more effectively than MTX. MTX1 blocks the Kv1.2 and Kv1.3 channels, expressed in Xenopus oocytes, with IC_50_ values of 0.26 and 180 nM, respectively. Moreover, MTX1 competitively interacts with both ^125^I-apamin (IC_50_ = 1.7 nM) and ^125^I-charybdotoxin (IC_50_ = 5 nM) for binding to rat brain synaptosomes. Despite its high sequence similarity (85%) to MTX, MTX1 exhibits a higher binding affinity towards the Kv1.2 and SKCa channels. Computational analysis highlights the significance of specific residues in the β-sheet region, particularly the R27, in enhancing the binding affinity of MTX1 towards the Kv1.2 and SKCa channels.

## 1. Introduction

Potassium channels are transmembrane proteins that play an essential role in the generation of electrical signals in the cell, underlying major physiological phenomena [1] such as neuronal communication [2] and muscle contraction [3], blood pressure regulation [4,5], cell proliferation and hormone secretion [6,7].

Scorpion venom neurotoxins have been shown to be highly selective and potent ligands for a wide range of ion channels and membrane receptors. As a result, they are promising compounds for the development of new drugs, such as analgesics, antivirals and anti-cancer treatments as well as drugs against neurological disorders [8,9,10,11].

For instance, several research projects have focused on the use of scorpion neurotoxins as potential new treatments against cancers. In fact, chlorotoxin, purified from the venom of *Leiurus quinquestriatus* scorpion [9,12,13] has already been the subject of clinical trials, as it specifically targets the invasion and migration of glioblastoma cells. Charybdotoxin and iberiotoxin, two blockers of Ca^2+^ activated K^+^ channels, from venoms of *Leiurus quinquestriatus* and *Mesobuthus tamulus* scorpions, respectively, were able to inhibit glioma and melanoma cell proliferation and migration [12,14], respectively. Furthermore, margatoxin extracted from *Centruroides margaritatus* scorpion venom inhibited the proliferation of human lung adenocarcinoma cells [15], by blocking the Kv1.2 and Kv1.3 channels [16].

Scorpion venom toxins active on the K^+^ channels (KScTx) are subdivided into α-, β- and γ-KTX families. The α-KTx family is considered the largest KScTx family and is subdivided into 31 subfamilies based on differences in amino acid sequences [17,18,19]. This family represents short-chain peptides comprising 23 to 42 amino acid residues cross-linked by three or four disulfide bridges with a molecular mass of around 4.6 KDa [20]. It shares a common structural motif known as the cysteine-stabilized α/β scaffold [21]. 

The three-disulfide-bridged scorpion toxins, such as charybdotoxin (CHTX), exhibit C1–4, C2–C5 and C3–C6 half-cystine pairing patterns [21,22], while the four-disulfide-bridged ones of the α-KTX6 subfamily exhibit a C1–C5, C2–C6, C3–C7 and C4–C8 pairing pattern, except maurotoxin (MTX), which was purified and characterized by Kharrat et al. [23] and features an unconventional disulfide-bridge pattern with an atypical C1–C5, C2–C6, C3–C4 and C7–C8 organization. MTX shares a high sequence identity with the other members of α-KTX6 subfamily, such as HTX from *Hemiscorpius lepturus* (73.5%), Pi1 from *Pandinus imperator* (61%) and HsTx1 from *Heterometrus spinnifer* (61.8%) scorpions [12,21,24,25]. These toxins differ from MTX in the organization of the last two disulfide bridges, contributing to the spatial distribution of the key residues involved in ion channel recognition [21].

In fact, MTX was able to compete with ^125^I-labeled kaliotoxin for binding to rat brain synaptosomes with an IC_50_ of 0.2 nM [26]. MTX is also a potent blocker of the Kv1.2 channels, expressed in *Xenopus laevis* oocytes, with an IC_50_ value of 0.8 nM, while its affinity for other potassium channels, such as Kv1.1 and Kv1.3, was greater than 40 nM [23,27,28]. This contrasts with charybdotoxin (CHTX), which is more potent against channels Kv1.3 and Kv1.1 than Kv1.2 [12].

In addition, MTX affects the calcium-activated small-conductance potassium (SKca) channels, as demonstrated by inhibiting the binding of ^125^I-apamin to rat synaptosomes, with an IC_50_ of 5 nM. Apamin is an 18-amino acid peptide neurotoxin present in *Apis mellifera bee* venom that selectively blocks SK channels by obstructing the channel pore [29]. Despite the different studies of maurotoxin and its activities on different K+ channel subtypes, the structural elements involved in the interaction with each K+ channel subtype have not been fully and clearly identified, because of its specific folding compared to the other α-KTX6 subfamily toxins. In this study we aim to identify and study the activities of new MTX-like peptide(s) in order to provide data allowing us to elucidate the structure–activity relationship of the MTX-like peptides. For this purpose, we screened the venom of the scorpion *Scorpio maurus palmatus*, using specific antibodies developed against MTX, in order to isolate novel peptides structurally related to MTX, with different activities.

## 2. Results

### 2.1. MTX1 Toxin Purification

The fractionation of crude *Maurus palmatus* scorpion venom, using a Sephadex G50 column, resulted in seven fractions (Figure 1A). Subsequent testing in mice revealed that only fraction IV was toxic by intracerebroventricular (i.c.v.) route injection. This particular fraction was subjected to additional fractionation on a C18 reversed-phase HPLC column (Beckman Coulter PN 235332, Brea, CA, USA), leading to the identification of multiple components (Figure 1B). The toxic fractions F1 and F2, eluting at 14.31 and 14.62, respectively, underwent chromatography on a C18 column under similar conditions albeit with a reduced gradient (1–20% in 50 min), facilitating the purification of the two peptides MTX1 and MTX, eluted at 25.19 min and 41.3 min with 10.07% and 16.52% acetonitrile-TFA, respectively (Figure 1C).

### 2.2. Amino Acid Sequence of MTX1 and Sequence Analysis

The MTX1 amino acid sequence was determined through Automatic Edman degradation of 2 nmol of S-pyridyl methylated peptides. MTX1 consists of 34 amino acids, including eight cysteine residues located at positions 3, 9, 13, 19, 24, 29, 31 and 34 as shown in Figure 2A. This Figure shows that MTX1 displays a high level of identity when compared to MTX (85.3%) and presents five substitutions in its sequence (D8E, R14K, I25M, K27R and S28K). The phylogenetic tree represented in Figure 2B using the iTOL (The Interactive Tree Of Life) [30] server showed the existence of four distinct groups: a first large group including MTX and MTX1, with most of the toxins belonging to alpha KTx6 and a few members belonging to alpha KTx23; a second group including members of alpha KTx15, KTx16, KTx7 and KTx12; a third group comprising members of alpha KTx3; and a final group comprising members of alpha KTx1, KTx4 and KTx2. Figure 2B shows that MTX1 shares a 44.7% sequence identity with CHTX (a fourth group member). 

### 2.3. Neurotoxic Activity of MTX1

Different doses of MTX1 (4, 6, 8, 10, 15 and 20 ng) were administered intracerebroventricularly (i.c.v.). Six mice were used for each dose. The rodents perished after a duration of 10 min. The toxicity of MTX1 exhibited a dose-dependent correlation, with the maximum fatality rate occurring at a concentration of 20 ng. The median lethal dose (LD_50_) was determined to be 10 ng per mouse. This outcome indicates that MTX1 is approximately eight times more lethal than MTX (LD_50_ = 80 ng/20 g) in mice.

### 2.4. Cross-Antigenicity of MTX1 with MTX Toxin

The antigenic homology between MTX1 and MTX was tested by ELISA, using different dilutions of specific anti-MTX mice polyclonal antibodies. MTX1 was well recognized by anti-MTX antibodies. These results suggested that MTX1 and MTX belong to the same antigenic group of KTX6 toxins. MTX1 and CHTX can induce specific antibodies that are recognized by MTX. The highest level of recognition was obtained after the second injection of 15 µg peptide. The optimal dilution of immune serum that exhibited the highest binding affinity for MTX was determined to be 1/100,000. The serum was evaluated for its ability to recognize MTX1 and CHTX using ELISA. The IC_50_ of MTX was identical to that of MTX1 and CHTX, indicating perfect antibody recognition and, therefore, a similar structure to MTX, and indicating that the two toxins belong to the same antigenic group (Figure 3).

### 2.5. Effect of MTX1 on Rat Brain Synaptosomes

MTX1 was first evaluated for its capacity to displace ^125^I-CHTX from its site in rat brain synaptosomes, using competitive assays. The results demonstrated that MTX1 exhibited competitive behavior with ^125^I-CHTX, by inhibiting its binding, with an IC_50_ value of 2 nM. Conversely, under identical conditions, native CHTX displayed ^125^I-CHTX with an IC_50_ value of 50 pM. This result indicated that MTX1 partially interacted with the CHTX-binding site in rat synaptosomes (Figure 4A).

We then assessed the capacity of MTX1 to rival ^125^I-apamin to bind rat brain synaptosomes. Figure 4B shows that MTX1 induced the concentration-dependent inhibition of ^125^I-apamin binding, with an IC_50_ value of 1.7 nM. Under identical conditions, native Apamin displayed ^125^I-apamin with an IC_50_ value of 0.5 pM. These results suggest that MTX1 recognizes apamin receptor sites with a high affinity and binds partially with these sites (Figure 4B).

### 2.6. Electrophysiological Recordings

MTX1 was studied for its activity against the Kv1.2 and Kv1.3 voltage-gated potassium channel isoforms. MTX1 showed a concentration-dependent effect on the blocking of both channels. At the highest concentration of 2.5 nM, MTX1 inhibited 90% and 80% of Kv1.2 and Kv1.3 currents, respectively (Figure 5A,D). To evaluate the potency, concentration-response curves were constructed, giving IC_50_ values of 0.26 nM and 180 nM for Kv1.2 and Kv1.3 channels, respectively (Figure 5B,E) with a Hill coefficient of H. = 1.1 ± 0.1. Extracellular application of MTX1 at 2.5 nM markedly reduced outward Kv1.2 and Kv1.3 currents at all test voltages examined (Figure 5C,F), and the kinetics of inhibition showed a partial recovery after the washing.

### 2.7. Molecular Modeling of MTX1, Kv1.2, Kv1.3 and SK2 Potassium Channels

In order to anticipate the interaction between the MTX1 and its receptors (Kv1.2, KV1.3 and SK2), a molecular modeling study was conducted, in order to construct 3D structures of the peptide and the targeted K^+^ channels. In fact, the high-resolution of the crystal structure of the Kv1.2/2.1 paddle chimera channel has been used for many research studies because it presents the best resolution. Furthermore, when we compare the structures of the channels (Kv1.3 and SK2) determined by cryo-EM and the structures generated by homology modeling, we find that they share the same conformation (the RMSD value is less than 0.9 Å), showing that this channel template is appropriate for this study [31]. Upon thorough examination, our results indicated that the structure of MTX1 is stabilized by four disulfide bridges, displaying a sequence signature characteristic of the cysteine-stabilized α/β (CSα/β) motif in comparison to MTX and its analogs (Figure 6). The potassium channel receptor proteins Kv1.2, Kv1.3 and SK2 were simulated using conventional molecular modeling techniques and refined as described in the Materials and Methods Section.

### 2.8. Toxin–Channel Docking Study

To study how MTX1 might interact with ion channels, a molecular docking analysis was conducted. The first step consisted of the preparation of the docking assembly, including two PDB entries consisting of the receptor file (crystal structure of the chimeric paddle channel Kv1.2–Kv2.1 code PDB 2R9R) and the ligand file (Maurotoxin MTX code PDB 1TXM).

Computational docking experiments were conducted to explore the intricate molecular aspects of the binding process and to formulate a plausible model of the interaction of MTX1 (the ligand) and potassium channels Kv1.2, KV1.3 and SK2 (the receptors). By utilizing PPD analysis, numerous interactions of moderate potency, with distances ranging between 1.7 Å and 5.7 Å, were anticipated. Upon thorough examination, it was determined that eight interactions in MTX1-KV1.2, resembling the optimal complex, were implicated. The involvement of various other amino acids in the selectivity pore in the interactions with MTX1 was also evident. Indeed, the results indicated that S6, K14, N21, K23, K30 and Y32 participate in the interaction process, forming hydrogen bonds with Q156, D44, Q46, Y176, Q376 and G67 of the Kv1.2 receptor at distances of 4.2 Å, 5.6 Å, 2.1 Å, 2 Å, 1.9 Å and 1.7 Å, respectively. Notably, K7 and R27, situated in the beta sheet of MTX1, are implicated in the interaction, as they establish a salt bridge with D178 and D264 at distances of 2.7 Å and 1.7 Å, respectively. In the helix, only R14 participates in the interaction with the channel. On the other hand, the MTX1–Kv1.3 complex interface displayed interactions involving S6, K7, N21, K23, N26, R27 and Y32 with G67, D68, H400, Y286, S156, P153 and G397 at distances of 2.8 Å, 2.4 Å, 1.8 Å, 2.7 Å, 2.2 Å and 2.4 Å, respectively. The anticipated mode of interaction for the MTX1–SK2 complex unveiled the presence of the residues K7, N21, K23, N26, R27, K30 and Y32 interacting with D58, D132, Y250, D326, Q325, D229 and G154, respectively and exhibiting a moderate strength with distances of 1.7 Å and 2.5 Å (Figure 7).

## 3. Discussion

Scorpion venom contains a variety of biologically active components, including neurotoxins that have been used to study the properties of different ion channels, particularly potassium channels [12]. These neurotoxins have therapeutic potentials due to their selective interaction with different Kv channels subtypes and serve as models for the advancement of molecular therapeutics [12,26]. The alpha KTx toxins represent the most extensive KTx family, comprising 31 subfamilies. These toxins consist of short-chain peptides of 23–42 amino acid residues linked by three or four disulfide bridges [17].

In this study, we isolated and characterized a new short peptide, which we called maurotoxin1 (MTX1). The identification of this peptide was achieved by bioguided screening of the venom fractions of the Tunisian scorpion species, *Scorpio maurus palmatus*, by recognition of specific antibodies developed against MTX, previously isolated from the same scorpion venom by Kharrat et al. [26]. The peptide was purified by a combination of gel filtration chromatography and RP-HPLC. 

The sequence of MTX1, determined by Edman degradation, consists of 34 amino acid residues containing eight half-cystine residues. It shares the highest sequence identity (85.3%) with maurotoxin (MTX) and differs by only five amino acid residues: E8D, K14R, M25I, R27S and K28R. It shares sequence identity with the other toxins of the alpha-KTx6 subfamily, including HsTx1 (61.8%) [32], Pi1 (57.1%) and Pi7 (52.6%) [33]. These short-chain toxins, of 34 to 38 amino acid residues, cross-linked by four disulfide bridges, differ from MTX in their half-cystine pairings. In addition, MTX1 shares over 50% sequence identity with the other voltage-gated K+ channels toxins subfamilies, and 26–64% sequence identity with toxins active on SKCa channels. 

When tested on mice, MTX1 showed eight times more toxicity than MTX, with an LD_50_ of 0.01 μg/mouse. Our electrophysiological results show that MTX1 had the most important effect on Kv1.2 currents, with an IC_50_ of 0.26 nM. This value is three times lower than that of MTX, which displayed an IC_50_ of 0.8 nM. However, MTX1 blocks the Kv1.3 channels with a comparable effect to MTX, with an IC_50_ value of 180 nM. Furthermore, we found that MTX1 was active on the apamin-sensitive SK_Ca_ channels with a high affinity (IC_50_ = 1.7 nM) almost three times higher than that of MTX (IC_50_ = 5 nM).

The three-dimensional structure (3D) model of MTX1 showed that it adopts a classical cysteine-stabilized α/β folding and a conserved conformation similar to that of MTX, which adopts an unconventional pattern of disulfide bond pairing, C1–C5, C2–C6, C3–C4 and C7–C8 [23]. These data suggest that the half-cysteine pairing pattern for MTX1 may be of an unconventional type, identical to that of MTX. Indeed, the presence of residue G33 between C7 and C8 in the C-terminal sequence of MTX1 favors an MTX-type disulfide bridge arrangement [21]. The proposed interaction model of MTX1 with Kv1.2, predicted by molecular docking, suggests the presence of eight amino acid residues (S6, K7, K14, N21, K23, R27, K30 and Y32) that interact with the Kv1.2 channel, whereas MTX interacts with only five amino acid residues (N21, K23, I25, N26, K30 and Y32) with the same channel. Thus, our model is in accordance with the electrophysiology results showing that MTX1 was more active than MTX on the Kv1.2 channel. Furthermore, an in silico study of MTX1’s interaction with the SK2 channel, carried out for the first time by molecular docking, highlighted the role of residues located in the β-sheet region (K7, N21, K23, N26, R27, K30 and Y32), and principally, of the residue R27, which allowed the toxin to orient itself towards the SK2 channel. This study contributes to a better understanding of how MTX1 interacts with the SK2 channels. Previous structure–function studies have highlighted the importance of basic residues R6 and R13 in scyllatoxin (ScTX) purified from the venom of the scorpion *Leiurus quinquestriatus hebraeus*, and residues R13 and R14 in Apamin are located in the helical region of the SKCa channel binding [34].

On the other hand, the MTX and MTX1 toxins share 44,7% and 50% sequence identities with CHTX, respectively. In addition, the C-ter cluster region of MTX and MTX1 comprising the hydrophilic amino acid residues 23rd aa–32nd aa (KCINKSCKCY and KCMNRKCKCY) is homologous to that of CHTX 27th aa–36th aa (KCMNKKCRCY). This could explain the cross-antigenicity results of MTX1 and CHTX toxins with anti-MTX antibodies, as well as the ability of the toxins to compete with [^125^I] charybdotoxin on rat brain synaptosomes with a high affinity (IC_50_ = 2 nM). All these results indicate structural and functional homologies between these toxins. Indeed, the phylogenetic tree that we constructed based on the sequence alignment of all the reported alpha KTx peptides showed that the members with similar identities were clustered together (Figure 2B).

Previous studies have demonstrated the significance of the typical functional dyad for the high-affinity blockade of the Kv channels. The 3D structure analysis revealed that the distances between K27–Y36 (the functional dyad) in CHTX and K23–Y32 in MTX1 were identical in the order of 6.7 A. These residues probably constitute the functional dyad of MTX1. Indeed, Castle et al. [25] demonstrated that substituting lysine 23 or tyrosine 32 in MTX affected its ability to block the IK, SK and Kv1.2 channels. The substitution of lysine 27 with a neutral amino acid (leucine) in Bot33 explains the lack of the inhibition of Kv channels in oocytes [19].

Thus, our results shed light on the importance of the key residues K7, N21, K23 and Y32, which are located in the beta-sheet region. These residues may play an essential role in the interaction of MTX1 and MTX with three channels, Kv1.2, Kv1.3 and SK2. Additionally, the critical role of residue R27 in MTX1 (K27 of MTX) was responsible for the increased affinity of MTX1 over that of MTX on the Kv1.2 and SK2 channels. Furthermore, residue K28 of MTX1 (equivalent to S28 of MTX), although not directly involved in channel interactions, appears to play an important role in the interactions with surface channels. These two substitutions may induce a conformational change in the 3D structure of MTX1, conferring a more optimal orientation to the K^+^ channels. In contrast, residue K23, with the same side-chain orientation, obstructed the selectivity pore for the Kv1.2, Kv1.3 and SK2 channels.

Our deep bioinformatics analysis allowed us to identify the structural elements responsible for the high affinity of MTX1 for potassium channel subtypes. This will allow us, in the future, to make modifications in order to improve the specificity of this toxin on a single SKCa or Kv1.2 channel subtype, allowing the design of more specific and non-toxic molecules for potential therapeutic use.

## 4. Materials and Methods

### 4.1. Scorpion Venom

The venom of the scorpion *Scorpio maurus palmatus*, obtained from Beni Khedach (Medenine, Tunisia), was electrically stimulated in the abdomen by a veterinarian at the Pasteur Institute of Tunisia. The venom sample (50 mg/mL) was stored at −20 °C in its natural state until needed.

### 4.2. Purification of MTX1

MTX1 toxin was isolated from raw venom through a two-stage purification process involving gel filtration, followed by reverse-phase HPLC. The mixture was clarified by centrifugation at 11,000× *g* (rpm) for 15 min at 4 °C. The resulting supernatant was then applied to a Sephadex G-50 column (1.7 × 150 cm) that had been pre-equilibrated with 0.1 M ammonium acetate buffer at pH 8.2. The eluted proteins were then monitored at 280 nm. The biologically active fraction (IV) was further separated using reverse-phase HPLC with a C18 column (5 µm, 4.6 250 mm, Beckman Coulter PN 235332, Brea, CA, USA). Elution was performed with a linear gradient of 0.1% trifluoroacetic acid in acetonitrile (5–25% for 40 min) at a flow rate of 1 mL/min. Fractions containing the primary toxic components were isolated using a C18 reversed-phase HPLC column (5 µm, 4.6 250 mm, Beckman Coulter) under similar conditions, with a linear gradient of 0.1% trifluoroacetic acid in acetonitrile (1% to 20% for 50 min) at a flow rate of 1 mL/min. The detection was performed at 215 nm.

### 4.3. Amino Acid Analysis and Sequence of MTX1

The reduction and alkylation of proteins and sequencing of native and S-alkylated peptides were performed according to previous protocols [35]. Automatic Edman degradation of 1 nmol native MTX1 displayed a consistent yield of 95% over the initial 31 cycles. PTH levels were comparatively low during these processes. For comprehensive sequence determination, the use of 1 nmol of S-alkylated peptide proved particularly beneficial in identifying cysteine positions and completing the MTX1 sequence.

### 4.4. Mass Spectrometry Analysis

MTX1 was analyzed using a voyager DE-RP matrix-assisted laser desorption/ionization time-of-flight (MALDI-TOF) mass spectrometer (Perspective Biosystems Inc., Framingham, MA, USA). The sample was dissolved in CH_3_CN/H_2_O (30/70) with 0.3% trifluoroacetic acid to obtain a concentration of 1–10 pmol·µL^−1^. The matrix solution was prepared by dissolving alpha-cyano-hydroxycinnamic acid in 50% CH_3_CN and 0.3% trifluoroacetic acid/H_2_O to form a saturated solution (10%)·µL^−1^. A 0.5 µL portion of the peptide solution was placed on the sample plate and 0.5 µL of the matrix solution was then added. The mixture was then allowed to dry. Mass spectra were obtained in linear mode, calibrated externally using appropriate standards and analyzed utilizing the GRAMS/386 Software (Galactic Industries Corporation and the Savitzky–Golay algorithm, version 6.00).

### 4.5. The Neurotoxic Activity of MTX1 in Mice

To assess the LD_50_ of the MTX1, different doses of the peptide were administered to males C57BL/6 mice (20 ± 2 g) by an intracerebroventricular injection under ether anesthesia [36], which is considered the most sensitive route for scorpion toxins in mammals. Each dose was suspended/diluted in 5 μL of 0.1% (*w*/*v*) BSA and tested on six mice. Symptoms of toxicity were observed and recorded over a 24 h period. The median lethal dose (LD_50_) corresponds to the dose required to kill half of the tested mice.

### 4.6. Mice Immunization

Upon retrieval of pre-immune serum via retro-orbital puncture, 30 µg of MTX1 or saline (0.9% NaCl) in complete Freund’s adjuvant was subcutaneously administered to 6-week-old mice on day 1. Subsequent booster injections of incomplete Freund’s adjuvant were administered 15 and 25 days later. On day 8, the mice were bled and used for ELISA experiments in 96-well microtitration plates (catalog number M5785-1CS Sigma Aldrich, MO, USA). The specific antibodies induced by MTX were distinctly recognized by MTX1, with the highest level of recognition achieved after the second injection of 15 µg peptide. The optimal dilution of the immune serum resulting in half-maximum binding of MTX to its specific antiserum was determined to be 1/100,000.

### 4.7. ELISA Assays

The specificity of anti-MTX against MTX1 and CHTX toxins was assessed using ELISA. One hundred microliters of 5 μg/mL of antigen (MTX1 and CHTX) in 0.1 M sodium bicarbonate buffer at pH 9.6 were adsorbed on 96-well NUNC plates for 90 min at 37 °C. After five washes with PBS/Tween (0.05%), non-specific sites were saturated with 0.5% BSA in PBS buffer for 1 h at 37 °C, followed by three washes with PBS/Tween (0.05%). Subsequently, appropriate serum dilutions were added and the samples were incubated at 37 °C for 90 min and 4 °C for 15 min. After washing with PBS/Tween (0.05%), 100 μL of peroxidase conjugate (anti-mouse diluted 1000-fold) was added to each well and incubated at 37 °C for 90 min and 4 °C for 15 min. Finally, 200 μL of 0.4 mg/mL OPD in citrate buffer (pH 5.2 containing H_2_O_2_ (0.03%) was added to each well, followed by a 5 min incubation in the dark at room temperature. The reaction was halted by adding 50 μL of 2 N sulfuric acid. The absorbance at 492 nm was measured using a spectrophotometer (Thermo-Multiskan EX, Shanghai, China). The immune serum titer corresponded to the dilution that yielded 50% of the maximum absorbance.

### 4.8. Competition Binding Assay on Rat Brain Synaptosomes and Iodination of ^125^I-MTX1

The rats were euthanized by dislocation and their brains were rapidly removed and placed on a cold plate. The brain tissue was suspended in a 10% (*w*/*v*) sucrose-HEPES buffer in a polytron homogenizer with a protease inhibitor. Synaptosomal membranes were prepared as previously described [37]. The protein content of the membranes, determined by the Bradford method, was 0.92 µg/µL. The protein content was determined using a modified Lowry method, with assays conducted on the synaptosomal fraction P2 isolated from rat brain. MTX1 was reacted with Na [^125^I] (Amersham Biosciences, Buckinghamshire, United Kingdom) in the presence of lactoperoxidase and H_2_O_2_ in 50 mM phosphate buffer at pH 7.2. The separation of free iodine was achieved by anion exchange chromatography using Dowex 1X-8 (Merck, Darmstadt, Germany). The mixture was then injected into a reverse-phase HPLC column (analytical RP-C18 Beckman, Ultrasphere, 5 μm, 4.6 × 250 mm). Solvent A consisted of 0.1% trifluoroacetic acid (*v*/*v*) in water, and solvent B consisted of 0.1% trifluoroacetic acid (*v*/*v*) in acetonitrile. The HPLC experiments were performed as previously described. Radioactivity was quantified using a Packard spectrometer (Crystal I1 multidetector system, San Francisco, CA, USA) with the native toxin eluted at 18 min. The iodination mixture yielded only two peaks (RT = 2 min and RdT = 25 min); approximately 70–85% of the radioactivity was linked to the peak (RT = 25 min), which was postulated to correspond to the mono-iod derivative.

### 4.9. Binding Assay for Kv Channels

A volume 400 µL of synaptosome suspension (0.4 mg protein/mL) in a solution of 50 mM Tris/HCl at pH 7.4 was prepared. MTX1 was introduced at different concentrations (ranging from 10–5 to 10–14 M) along with 40 pM ^125^I-CHTX diluted in a mixture of 100 mM NaCl, 20 mM Tris/HCl at pH 7.4, and a variety of concentrations of MTX1, MTX and CHTx were prepared. The binding process was allowed to proceed at 25 °C for 30 min in an incubation buffer composed of 50 mM NaCl, 20 mM Tris/HCl and 0.1% bovine albumin at pH 7.4. The reaction was terminated by adding ice-cold 100 mM NaCl and 20 mM Tris/HEPES (pH 7.4, 1 mL), followed by centrifugation (10 min). The resultant pellets were washed three times in 1 mL of identical buffer, and the correlated radioactivity was quantified using a γ counter (Packard Crystal II). Each concentration of MTX1 was assessed in triplicates.

### 4.10. Binding Assay for SKCa Channels

The process of iodination of apamin was executed utilizing IODO-GEN. In displacement assays, 50 μL of 0.1 nM ^125^I-apamine (2000 Ci/mmol) was combined with 400 µL of synaptosome suspension (0.4 mg protein/mL). The samples were incubated with 50 µL of one of the competitor concentration series (MTX1, MTX or CHTX) in a final volume of 500 µL for 1 h at 0 °C. The incubation buffer consisted of 25 mM hydroxymethyl amino methane (Tris-HCl) (pH 7.2), 10 mM potassium chloride (KCl) and 0.1% BS. Following centrifugation for 10 min, the samples were centrifuged again and the resulting pellets were washed three times with 2 mL of the same buffer. The quantification of bound radioactivity was conducted utilizing (Packard Crystal II). The mean values of triplicate experiments are reported. Non-specific binding, characterized as less than 10% of total binding, was determined in the presence of an excess (10 nM) of unlabeled apamin.

### 4.11. Oocyte Preparation and Electrophysiological Recordings

The *xenopus* was anesthetized in a bath of tricaine solution. It was then placed on its back in a dissecting tank covered with alcohol-cleaned BenchGuard paper. A horizontal slit was made in the animal’s lower abdomen by incising a piece of the underlying muscle with the skin. The ootheca were then gently pulled outwards and a portion was cut out and transferred to a sterile Erlen tube containing MBS medium, leaving the rest in place in the ovary. Finally, the incision was closed using sterile, absorbable sutures.

*Xenopus laevis* oocytes at stages V and VI were retrieved and prepared for cRNA injection and electrophysiological recordings. The enzymatic elimination of oocyte follicular cell layers was carried out by employing 2 mg·mL^−1^ collagenase IA (Sigma, St. Louis, MO, USA) in the classical Barth’s medium containing (in mm) 88 NaCl, 1 KCl, 0.82 MgSO_4_, 0.33 Ca(NO_3_)_2_, 0.41 CaCl_2_, 2.4 NaHCO_3_, and 15N-2-hydroxyethylpiperazine-N′-ethanesulfonic acid (Hepes), with a pH of 7.4 (NaOH). The plasmids were digested using Xba1 (rat Kv1.2) and EcoR1 (rat Kv1.3). The resultant linearized plasmids were transcribed using either a T7 or SP6 mMessage mMachine transcription kit (Ambion, Austin, TX, USA). Subsequently, cRNAs (1 µg·µL^−1^) were cryopreserved in H_2_O at −80 °C. Injection of cells with 50 nL of cRNA (0.2 µg·µL^−1^ rat Kv1.2, or Kv1.3 channels) took place 2 days later. To promote the expression of ion channels, the cells were cultured at 16 °C in a specified nutrient oocyte medium for 2–6 days prior to conducting current recordings. Oocyte currents were captured at 20–22 °C using conventional two-microelectrode techniques with a voltage-clamp amplifier (GeneClamp 500; Axon Instruments, Foster City, CA, USA) linked to a 16-bit AD/DA converter (Digidata 1200A, Axon Instruments). The electrodes filled with 140 mM KCl exhibited an electrical resistance of 0.5–1 MΩ. Voltage pulses were administered every 15 s, maintaining a holding potential of −80 mV. Current data were sampled at 10 kHz, subjected to low-pass filtration at 2 kHz through an eight-pole Bessel filter and archived on a computer for subsequent assessment. The extracellular solution for recording contained (in mM) 88 NaCl, 10 KCl, 2 MgCl_2_, 0.5 CaCl_2_, 0.5 niflumic acid, 5 Hepes, and 0.1% bovine serum albumin, with a pH of 7.4 (NaOH). The P/4 protocol was implemented for online subtraction of the leak and capacitive currents. Any residual capacitive artifacts were eliminated for visualization purposes. The peptide solutions of MTX1 were perfused into the recording chamber at a flow rate of 2 mL/min using a ValveBank4 apparatus (Automate Scientific Inc., St Berkeley, CA 94710, USA). Data analysis was conducted using the software patch clamp 6.0.3 (Axon Instruments, Foster City, CA, USA). The outcomes are presented as the mean ± SEM.

### 4.12. Computational Study

#### 4.12.1. Phylogenetic Tree Determination

The entire analysis was performed using a server operating on Linux—Ubuntu 32.04.3 TLS with 32 GB RAM allocated for memory. All alpha-KTx members were identified through a UniProt sequence search that was executed in the current study, followed by a thorough analysis (phylogenetic analysis). Utilizing the protein basic local alignment search tool (BLAST: https://blast.ncbi.nlm.nih.gov/Blast.cgi, accessed on 15 November 2023), homologous sequences to MTX and MTX1 were retrieved. Specifically, non-redundant UniProtKB/SwissProt sequences were chosen as the subject database [38], and the Position-Specific Iterated BLAST (PSI-Blast) algorithm was employed for sequence alignment. To identify shared functional patterns, the MSA of toxins was determined using MAFFT (Multiple Alignment using Fast Fourier Transform) v7.490 [39]. The evolutionary history was inferred using the neighbor-joining technique [40]. Evolutionary distances were calculated using the JTT matrix-based method and expressed in terms of the number of amino acid substitutions per site [41]. A gamma distribution (shape parameter = 1) was used to depict the rate of variance among sites. The evolutionary analyses were performed using MEGA X [42].

#### 4.12.2. Molecular Modeling of MTX1, Kv1.2, Kv1.3 and SKCa2.2 Potassium Channel

The alignment of target sequences with templates from the Protein Data Bank (PDB) was achieved using the Needleman–Wunsch algorithm in EMBOSS [43]. Comparative modeling based on the satisfaction of spatial restraints was conducted using MODELLER version 9.24 [44]. A model for the Kv1.3 and SK2.2 channel structure was constructed based on homology to the crystal structure of the Kv1.2–Kv2.1 paddle chimera channel. The human Kv1.3 potassium channel subtype sequence, comprising 511 amino acids, was obtained from the Uniprot database under accession number P48547. The template chosen for modeling MTX1 was maurotoxin (MTX) sourced from *Scorpio maurus* (PDB code 1TXM). Models with the highest DOPE scores were selected after generating 1000 conformers [45]. The final 3D modeled structure was validated using the Ramachandran plot analysis (PROCHECK) by Laskowski et al. [46] and Verify 3D by Eisenberg et al. [47].

#### 4.12.3. Toxin–Channel Docking Study

Protein–protein docking was performed to study the interaction between the ligand (MTX and MTX1) and receptor (Kv1.2, Kv1.3, and SKCa2.2) via the selectivity pore, as identified in the previous network based on protein–domain interactions [48]. ClusPro 2.0software (https://cluspro.org/, accessed on 8 July 2024) was used to perform the protein–protein blind docking [49]. ClusPro represents the best docking server, as demonstrated in CAPRI rounds over the last 20 years. All models of the docking ensemble were generated and selected based on several evaluation criteria. In this step, we minimized all structures before running the deep molecular docking study [50]. After the analysis and visualization of the results, the retained complex corresponded to the best interaction mode. All 3D structures were visually explored using PyMOL 3.0 [51] molecular viewer (version 1.7.2.1).

## 5. Conclusions

In this study, we isolated MTX1, a new highly toxic peptide from the venom of the Tunisian scorpion *Scorpio maurus palmatus.* It shares a high structural similarity with maurotoxin (MTX) and other toxins of the alpha-KTx6 family but exhibits some differences that translate into interesting functional properties. While MTX1 blocks the Kv1.3 potassium channels in a comparable manner to MTX, it is significantly more effective at inhibiting the Kv1.2 currents and SKCa channels. Interestingly, MTX1 also shares similarities with charybdotoxin (CHTX), a toxin with different properties. This suggests that slight variations in sequence and structure enable these scorpion peptides to target a wide variety of ion channels, with remarkable affinity and selectivity. Molecular modeling studies have allowed us to identify the amino acid residues essential for the binding of MTX1 to target channels, including, in particular, residue R27, which appears to be responsible for MTX1’s increased affinity for the Kv1.2 and SK2 potassium channels. Thus, structural bioinformatics has become essential for deciphering the interactions between toxins and their therapeutic targets, and the search for new selective toxins will enable the design and development of new pharmacological agents capable of modulating ion channel functions.

## Figures and Tables

**Figure 1 ijms-25-10472-f001:**
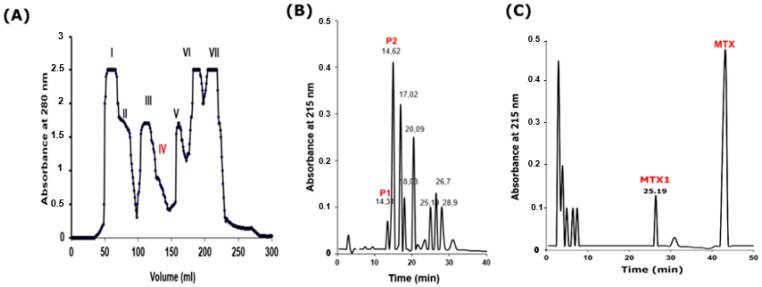
Purification of MTX1 from *Scorpio maurus palmatus* scorpion venom. (**A**) Chromatography of Sephadex G-50 column. Fractions I–VII were collected. (**B**) Chromatography of fraction IV by using C18 RP-HPLC. (**C**) Chromatography of pics P1 and P2 eluted at 14.31 min and 14.62 min on C18-RP-HPLC. MTX1 and MTX were collected at 25.19 min and 41.3 min, respectively.

**Figure 2 ijms-25-10472-f002:**
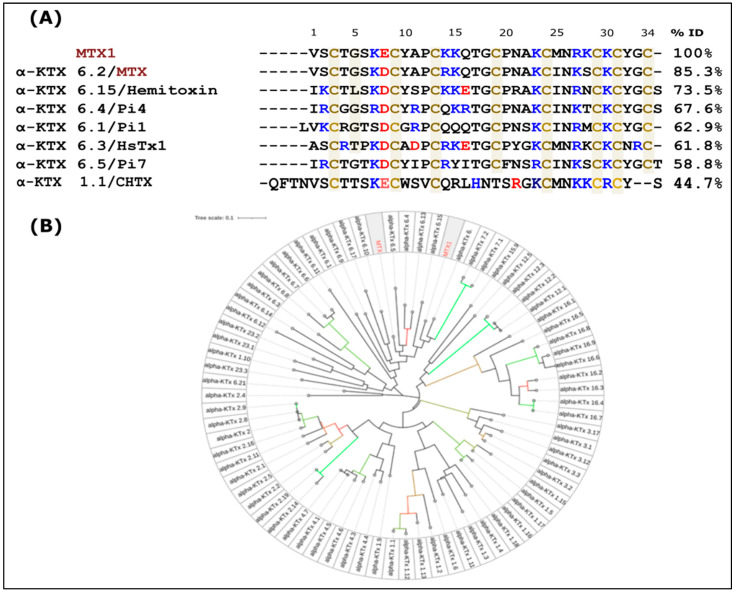
(**A**) The sequence of MTX1 aligned with Kv scorpion toxins. The MTX1 amino acid sequence was revealed by automatic Edman degradation. The alignment of MTX1 with members of the alpha-KTx subfamily is shown. Cysteine residues are highlighted in green. The numbers next to each toxin name represent the percentage identity (ID) of MTX1. (**B**) A phylogenetic tree depicting the relationship between MTX, MTX1 and the alpha-KTx toxin family is shown. Phylogenetic analyses were conducted using the JTT matrix-based method, considering the sequence homology of all the toxins. This analysis encompassed 34 to 38 amino acid sequences. The topology of the tree was confirmed using 1000 bootstrap replicates. Bootstraps around 70 are highlighted in red, whereas those around 100 are shown in green.

**Figure 3 ijms-25-10472-f003:**
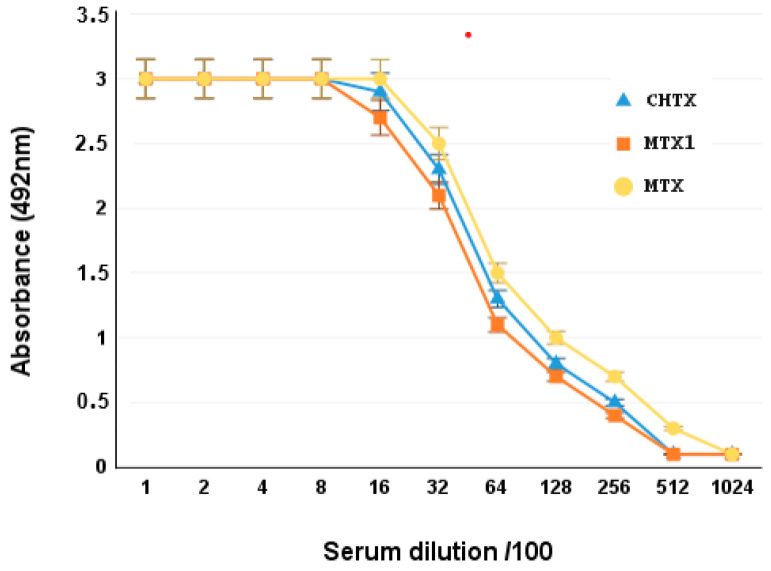
Cross-antigenicity characterization of MTX1, MTX and CHTX. Binding of anti-MTX serum to plates coated with 5 µg/mL of MTX (●), MTX1 (■) or CHTX (▲).

**Figure 4 ijms-25-10472-f004:**
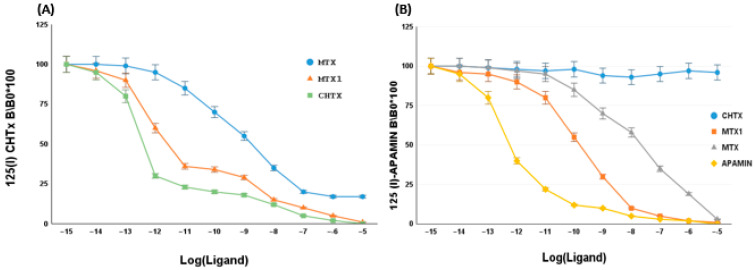
Competitive binding on rat brain synaptosomes of MTX1 and potassium channel toxins. (**A**) Inhibition of ^125^I-CHTX binding to rat brain synaptosomes by unlabeled CHTX (▪), MTX1 (▲) and MTX (●). Instances of non-specific binding, defined as less than 20% of total binding of ^125^I-CHTX, were subtracted from the ratio calculation. (**B**) Inhibition of ^125^I-apamin binding to rat brain synaptosomes by unlabeled Apamin (⧫), MTX (▲) and MTX1 (■) and CHTX (●). Instances of non-specific binding, defined as less than 10% of total binding of ^125^I-apamin, were subtracted from the ratio calculation. BO represents radiolabeled toxins binding in the absence of a ligand, and B represents binding in the presence of the indicated concentrations of competitors. The values shown on the curves represent the average of three experimental values. The errors less than or equal to 0.05.

**Figure 5 ijms-25-10472-f005:**
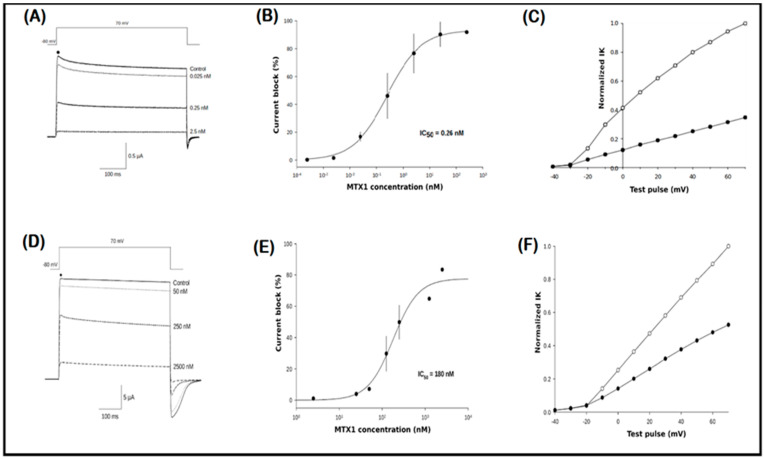
The blocking efficacy of MTX1 on the Kv1.2 and Kv1.3 channels. (**A**,**D**) Oocytes expressing the Kv1.2 or Kv1.3 channel were recorded using a two-electrode voltage clamp. K+ currents were obtained by depolarization from a holding potential of −80 mV to +70 mV under different concentrations of MTX1 toxin, illustrating Kv1 channel block. (**B**,**E**). Dose-response effects of MTX1 on Kv1.2 and Kv1.3 channel currents with IC_50_ values of 0.26 nM and 180 nM, respectively, with a Hill coefficient of 1.1 ± 0.1. (**C**,**F**) Comparison of K^+^ control current (○) and inhibition by MTX1 at 2.5 µM for Kv1.2 and 2.5 nM for Kv1.3 (●). Data points represent the mean ± SEM. The tail current likely consists of a combination of inward K^+^ current and outward Cl^−^ current. The addition of niflumic acid aimed to inhibit chloride currents, yet its efficacy remains incomplete. In the case of the inward potassium current, it is plausible that, at a holding potential of −80 mV, we surpass the ion’s reversal potential, possibly accounting for the inward current during repolarization and the toxin’s partial impact. Reduction in tail current mirrored the ionic current’s amplitude, indicating the toxin’s specific action on the channel.

**Figure 6 ijms-25-10472-f006:**
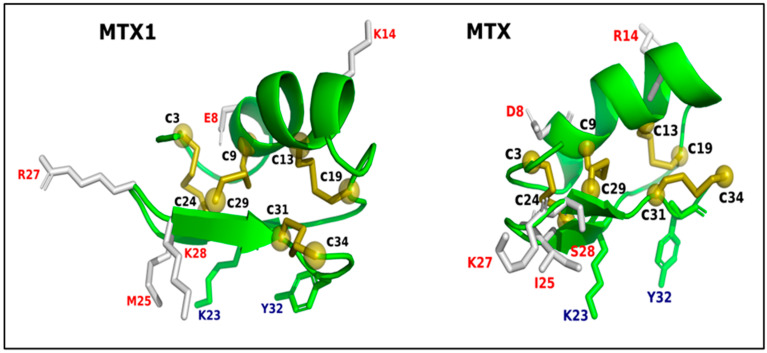
A cartoon representation of the 3D structures of the MTX1 and MTX. The hotspots at positions 23 and 32 are represented as sticks, the disulfide bridges are represented in spheres and colored olive and the different amino acids between MTX1 and MTX are represented by white colored sticks.

**Figure 7 ijms-25-10472-f007:**
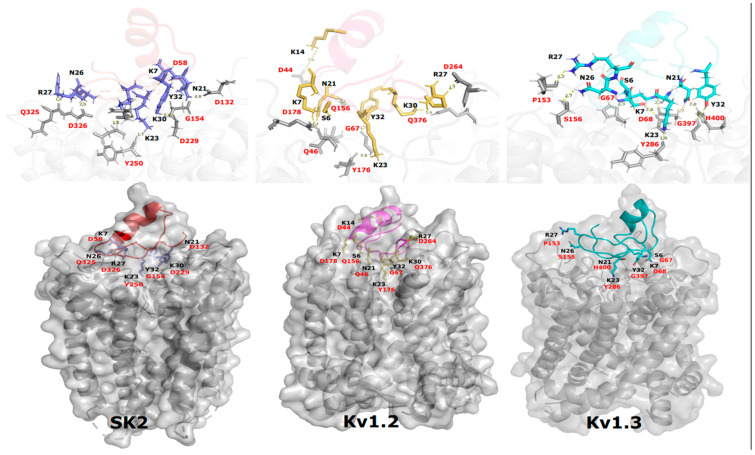
Interaction modes of MTX1 with SK2, Kv1.2 and KV1.3 as predicted by a protein–protein docking study. Residues defining the toxin–channel interactions are highlighted in black for the ligand (MTX1) and in red for the targeted channel. Elementary interactions are shown in zoom.

## Data Availability

Data available on request from the authors.

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
