# Peer review of "Structure–Function Relationship of a Novel MTX-like Peptide (MTX1) Isolated and Characterized from the Venom of the Scorpion Maurus palmatus"

_ijms, 2024, doi:10.3390/ijms251910472_

Round 1

Reviewer 1 Report

Comments and Suggestions for Authors

El Fessi and co-workers describe the identification, purification, and initial characterization of a novel component of the venom of Maurus palmatus closely related to maurotoxin (MTX) that they name MTX1. The novel peptide was purified by sequential RP chromatography (with increasing resolution) of a size exclusion fraction that also contained MTX. Edman degradation sequencing revealed five differences between MTX and MTX1. MTX1 contains eight cysteine residues, which allows assigning it to the four-disulfide bridges group of toxins (like MTX). MTX1 was neurotoxic, as intraventricular injection led to the death of mice within 10 minutes with an LD50 of 10ng. MTX antibodies show cross-reactivity with MTX1 and CHTX, supporting a close structural conservation between toxins.

In an attempt to get closer information on potential binding sites, competition determinations with CHTX and apamin were performed. In both cases, MTX1 displaced the ligand with higher affinity than MTX, indicating that the binding sites in rat synaptosomes partially overlap.

The inhibitory activity of MTX1 on Kv1.2 and Kv1.3 was tested using Xenopus oocytes. In both cases, the authors report dose-dependent inhibition. The study then approaches the computational modeling of the binding of MTX1 to Kv1.2, Kv1.3, and SK2 channels and compares its features with MTX1. This gives insights into the residues crucial for the interaction.

There are some points that would need further clarification.

-A more thorough description of the modeling of the channels would be helpful. Using the crystal structure of Kv1.2/2.1 chimera is a very reasonable starting point, but there are other structures available (for both SK2 and Kv1.3), albeit with cryo-EM resolution. Were these taken into account?

-In Fig. 5 A and D, the tail current at the end of the stimulus is very different. The large tail in D also seems to be reduced in the presence of toxin. Can the authors comment on that?

Minor points

-Figure 4. Errors “around 5%” is a vague description; why are error bars not depicted?

-Figure 5 E and F are not described in the legend.

Typos and trivial errors.

Page 7, Line 182, “methods” come afterwards, not previously.

Page 12, Line 417: -80mV?; Line 420: Niflumic acid?

Author Response

Structure-function relationship of a novel MTX-like peptide (MTX1) isolated and characterized from the venom of the scorpion Maurus palmatus. ID: ijms-3079363

Response to Reviewer 1 Comments

1. Summary

El Fessi and co-workers describe the identification, purification, and initial characterization of a novel component of the venom of Maurus palmatus closely related to maurotoxin (MTX) that they name MTX1. The novel peptide was purified by sequential RP chromatography (with increasing resolution) of a size exclusion fraction that also contained MTX. Edman degradation sequencing revealed five differences between MTX and MTX1. MTX1 contains eight cysteine residues, which allows assigning it to the four-disulfide bridges group of toxins (like MTX). MTX1 was neurotoxic, as intraventricular injection led to the death of mice within 10 minutes with an LD50 of 10 ng. MTX antibodies show cross-reactivity with MTX1 and CHTX, supporting a close structural conservation between toxins.

In an attempt to get closer information on potential binding sites, competition determinations with CHTX and apamin were performed. In both cases, MTX1 displaced the ligand with higher affinity than MTX, indicating that the binding sites in rat synaptosomes partially overlap.The inhibitory activity of MTX1 on Kv1.2 and Kv1.3 was tested using Xenopus oocytes. In both cases, the authors report dose-dependent inhibition. The study then approaches the computational modeling of the binding of MTX1 to Kv1.2, Kv1.3, and SK2 channels and compares its features with MTX1. This gives insights into the residues crucial for the interaction.

There are some points that would need further clarification.

2. Point-by-point response to Comments and Suggestions for Authors

Thank you very much for taking the time to review this manuscript. Please find the detailed responses below and the corresponding revisions in track changes in the re-submitted files.

Comments 1: A more thorough description of the modeling of the channels would be helpful. Using the crystal structure of Kv1.2/2.1 chimera is a very reasonable starting point, but there are other structures available (for both SK2 and Kv1.3), albeit with cryo-EM resolution. Were these taken into account?

Response1:

We agree with the reviewer on this issue. Therefore, we have modified the paragraph to improve clarity in Section 2.7. Molecular modeling of MTX1, Kv1.2, Kv1.3, and SK2 potassium channels of the actual version: “In fact, the high-resolution of the crystal structure of the Kv1.2/2.1 paddle chimera channel has been used for many research studies because it presents the best resolution. Furthermore, when we compare the structures of the channels (of the Kv1.3 and SK2) determined by cryo-EM and the structures generated by homology modeling, we find that they share the same conformation (the RMSD value is less than 0.9 Å), showing that this channel template is appropriate for this study.

Comments 2: In Fig. 5 A and D, the tail current at the end of the stimulus is very different. The large tail in D also seems to be reduced in the presence of toxin. Can the authors comment on that?

Response 2:

The tail current likely consists of a combination of inward K+ current and outward Cl- current. The addition of niflumic acid aimed to inhibit chloride currents, yet its efficacy remains incomplete. In the case of the inward potassium current, it is plausible that, at a holding potential of -80 mV, we surpass the ion's reversal potential, possibly accounting for the inward current during repolarization and the toxin's partial impact. Reduction in tail current mirrored the ionic current's amplitude, indicating the toxin's specific action on the channel. We indicate this fact now in the legend of the figure.

Minor points

Point 1: Figure 4. Errors “around 5%” is a vague description; why are error bars not depicted?

Response point 1:

The authors agree with the reviewer. Accordingly, we have revised the figure 4 by adding error bars and modified the legend by replacing "around 5%" with "less than or equal to 0.05”. These adjustments were made to provide clearer and more precise information.

Point 2: Figure 5 E and F are not described in the legend.

Response 2: The reviewer is right. Accordingly, we have thus revised the legend and included a description for Figure 5 E and F on page 7 from lines 195 to 204.

Point 3: Page 7, Line 182, “methods” come afterwards, not previously

Response 3: We totally agree with the comment made by the reviewer, this is now rephrased on page 7 Line 220.

Point 4: Page 12, Line 417: -80mV? Line 420: Niflumic acid?

Response 4: Thank you for pointing out these mistakes. We replaced 80 mv with -80mV on page 12, Line 488 and niflumic acids with niflumic acid at line 491.

4. Response to Comments on the Quality of English Language

Reviewer response: English language fine. No issues detected

5. Additional clarifications

Nothing to add

Response to Reviewer 2 Comments

Reviewer 2 Report

Comments and Suggestions for Authors

El Fessi et al. present a prospective investigation wherein a scorpion venom toxin, MTX, was to be isolated and purified to assess binding characteristics with SKca channels. However, the authors serendipitously found another, similar toxin and named it MTX1. Thereafter, this new toxin was characterized in terms of amino acid composition and biological activity against intact animals and cell culture. Modeling of docking of the new toxin with three channels was also performed. The rationale for the investigation was that finding such toxins in some way could be used in medicine to treat inflammatory diseases, cancer, etc. There are many issues to be addressed.

Introduction.

Rationale – The authors mention potential benefits to treat disease but provide no information on how such toxins would be of any benefit. The data subsequently presented demonstrates the lethal potency, not clinical utility, of these toxins. Please address this problem.

Hypothesis - The authors need to better state what their hypothesis was or rationale for the study. Lines 70-75 provide some insight, but as written, the authors wanted to isolate MTX and identify the molecular site with which this peptide interacts with SKca to block the channel. In the process of isolation, it sounds as if the authors found not one, but two fractions of the venom that caused toxicity, and isolation and characterization of the two fractions revealed two similar peptides with very different activities – one being MTX based on known amino acid sequence, and the other peptide now named MTX1. The find is serendipitous, important, but not well introduced. Please modify this part of the Introduction accordingly.

Results

While reasonably organized and presented, not much detail is provided to indicate many facts needed to assess the data. Number of replicate experiments, doses of venom used in vivo, mechanism of death, etc. are not provided. Please provide more detail in the various legends.

Methods.

Mouse experiments – How were the mice managed for the LD50 studies? Were they anesthetized, and how were the injections of venom performed? What were the doses used to generate the LD50 values? What software was used to analyze these data?

Rat experiments – How were the rats managed to obtain the tissues and cells obtained? Were they anesthetized?

Xenopus eggs – How were the toads handled to obtain the oocytes? Were they anesthetized?

Molecular modeling experiments – how many replicates for each concentration or condition were performed to obtain the results of the various relevant figures?

In summary, the rationale for the investigation is not clear and the potential benefit to molecular medicine is not presented. The handling of the animals involved is not presented. Please address these key issues.

Author Response

Response to Reviewer 2 Comments

1. Summary

El Fessi et al. present a prospective investigation wherein a scorpion venom toxin, MTX, was to be isolated and purified to assess binding characteristics with SKca channels. However, the authors serendipitously found another, similar toxin and named it MTX1. Thereafter, this new toxin was characterized in terms of amino acid composition and biological activity against intact animals and cell culture. Modeling of docking of the new toxin with three channels was also performed. The rationale for the investigation was that finding such toxins in some way could be used in medicine to treat inflammatory diseases, cancer, etc.

There are many issues to be addressed.

Thank you very much for taking the time to review this manuscript. Please find detailed responses below and corresponding revisions tracked as changes in the resubmitted files.

2. Point-by-point response to Comments and Suggestions for Authors

Comments 1: Introduction.

Rationale – The authors mention potential benefits to treat disease but provide no information on how such toxins would be of any benefit. The data subsequently presented demonstrates the lethal potency, not clinical utility, of these toxins. Please address this problem.

Response 1: The authors would like to thank the reviewer for pointing this out. Therefore, we have made the following changes to the manuscript in the introduction section (in blue color), by reporting some examples of scorpion toxins used as potential therapeutic molecules. Furthermore, a paragraph was added at the end of the discussion section in order to clarify the potential role of this toxin for the design of more specific and non-toxic molecules for potential therapeutic use.

Comments 2: Hypothesis - The authors need to better state what their hypothesis was or rationale for the study. Lines 70-75 provide some insight, but as written, the authors wanted to isolate MTX and identify the molecular site with which this peptide interacts with SKca to block the channel. In the process of isolation, it sounds as if the authors found not one, but two fractions of the venom that caused toxicity, and isolation and characterization of the two fractions revealed two similar peptides with very different activities – one being MTX based on known amino acid sequence, and the other peptide now named MTX1. The find is serendipitous, important, but not well introduced. Please modify this part of the Introduction accordingly.

Response 2: We agree with your comment. We think that we did not express ourselves well in the first version of our manuscript. We have therefore revised it to clarify this point in the end of the introduction section

“Despite the different studies of Maurotoxin and its activities on different K+ channel subtypes, the structural elements involved in the interaction with each K+ channel subtype have not been fully and clearly identified because of its specific folding, compared to the other α-KTX6 subfamily toxins.  In this study we aim to identify and study the activities of new MTX-like peptide(s) in order to provide data allowing us to elucidate the structure-activity relationship of the MTX-like peptides. For this purpose, we screened the venom of the scorpion Scorpio maurus palmatus, using specific antibodies developed against MTX, in order to isolate novel peptides structurally related to MTX, with different activities.”

Comments 3: Results- While reasonably organized and presented, not much detail is provided to indicate many facts needed to assess the data. Number of replicate experiments, doses of venom used in vivo, mechanism of death, etc. are not provided. Please provide more detail in the various legends.

Response 3:  We concur with the reviewer's observation, which has now been addressed. More details are now provided on page 4, specifically in paragraph "2.3. Neurotoxic activity of MTX1." Different doses of MTX1 (4, 6, 8, 10, 15, and 20 ng) were administered intracerebroventricularly (i.c.v.). Six mice were used for each dose. The rodents perished after a duration of 10 minutes. The toxicity of MTX1 exhibited a dose-dependent correlation, with the maximum fatality rate occurring at a concentration of 20 ng. The median lethal dose (LD50) was determined to be 10 ng per mouse. This outcome indicates that MTX1 is approximately eight times more lethal than MTX (LD50 = 80 ng/20 g) in mice.

Comments 4: Methods. Mouse experiments – How were the mice managed for the LD50 studies? Were they anesthetized, and how were the injections of venom performed? What were the doses used to generate the LD50 values? What software was used to analyze these data?

Response 4: These issues were addressed in the revised version of the manuscript on page 10, paragraph “4.5. The neurotoxic activity of MTX1 in mice.”To assess the LD50 of the MTX1,  different doses of the peptide were administered to males C57BL/6 mice (20 ± 2 g) by an intracerebroventricular injection under ether anesthesia (Galeotti et al. 2003), which is considered the most sensitive route for scorpion toxins in mammals. Each dose was suspended/diluted in 5 μL of 0.1% (w/v) BSA, and tested on six mice. Symptoms of toxicity were observed and recorded over a 24-hour period. The median lethal dose (LD50) corresponds to the dose required to kill half of the tested mice.

Comments 5: Rat experiments – How were the rats managed to obtain the tissues and cells obtained? Were they anesthetized?

Response 5: We have addressed your concerns in the revised version of the manuscript on page 10, paragraph “4.8. Competition Binding Assay on Rat Brain Synaptosomes and Iodination of 125I- MTX1”

Rat experiments – The rats were euthanized by dislocation and their brains were rapidly removed and placed on a cold plate. The brain tissue was suspended in a 10% (w/v) sucrose-HEPES buffer in a polytron homogenizer with a protease inhibitor. Synaptosomal membranes were prepared as previously described by Gray and Whittaker (1962). The protein content of the membranes, determined by the Bradford method, was 0.92 µg/µL.

Comments 6: Xenopus eggs – How were the toads handled to obtain the oocytes? Were they anesthetized?

Response 6: We provide further clarification on the method used for oocyte extraction in paragraph "4.11. Oocyte preparation and electrophysiological recordings”.

The xenopus was anesthetized in a bath of tricaine solution. It was then placed on its back in a dissecting tank covered with alcohol-cleaned BenchGuard paper. A horizontal slit was made in the animal's lower abdomen by incising a piece of the underlying muscle with the skin. The ootheca were then gently pulled outwards and a portion was cut out and transferred to a sterile Erlen tube containing MBS medium, leaving the rest in place in the ovary. Finally, the incision was closed using sterile, absorbable sutures.

Comments 7: Molecular modeling experiments – how many replicates for each concentration or condition were performed to obtain the results of the various relevant figures?

Response 7: Molecular modeling is an in silico technique used to predict the 3D structure of an unknown molecule.  We mentioned in Section 4.12.2. Molecular modeling of MTX1, Kv1.2, Kv1.3 and SKCa2.2 potassium channels that “Models with the highest DOPE scores were selected after generating 1000 conformers.”

Comments 8: In summary, the rationale for the investigation is not clear and the potential benefit to molecular medicine is not presented. The handling of the animals involved is not presented. Please address these key issues.

Response 8: We fully agree with the reviewer's comment. The manuscript has been revised accordingly in its different sections with blue color.

4. Response to Comments on the Quality of English Language

Reviewer response: English language fine. No issues detected

Round 2

Reviewer 2 Report

Comments and Suggestions for Authors

No further comments. My questions have been answered.